# A Consortium Blockchain-Based Agricultural Machinery Scheduling System

**DOI:** 10.3390/s20092643

**Published:** 2020-05-06

**Authors:** Haotian Yang, Shuming Xiong, Samuel Akwasi Frimpong, Mingzheng Zhang

**Affiliations:** 1School of Computer Science and Communication Engineering, Jiangsu University, Zhenjiang 212013, China; b342634705@163.com (H.Y.); zhangmz0117@163.com (M.Z.); 2School of Business, The University of Nottingham, Nottingham NG7 2RD, UK; 3Department of Computer Engineering, Ghana Technology University College, Accra PMB 100, Ghana; sfrimpong@gtuc.edu.gh

**Keywords:** consortium blockchain, matching function, scheduling algorithm, smart contract, supervision, consensus algorithm

## Abstract

The introduction of a consortium blockchain-based agricultural machinery scheduling system will help improve the transparency and efficiency of the data flow within the sector. Currently, the traditional agricultural machinery centralized scheduling systems suffer when there is a failure of the single point control system, and it also comes with high cost managing with little transparency, not leaving out the wastage of resources. This paper proposes a consortium blockchain-based agricultural machinery scheduling system for solving the problems of single point of failure, high-cost, low transparency, and waste of resources. The consortium blockchain-based system eliminates the central server in the traditional way, optimizes the matching function and scheduling algorithm in the smart contract, and improves the scheduling efficiency. The data in the system can be traced, which increases transparency and improves the efficiency of decision-making in the process of scheduling. In addition, this system adopts a crowdsourcing scheduling mode, making full use of idle agricultural machinery in the society, which can effectively solve the problem of resource waste. Then, the proposed system implements authentication access mechanisms, and allows only authorized users into the system. It includes transactions based on digital currency and eliminates third-party platform to charge service fees. Moreover, participating organizations have the opportunity to obtain benefits and reduce transaction costs. Finally, the upper layers supervision improves the efficiency and security of consensus algorithm, allows supervisors to block users with malicious motives, and always ensures system security.

## 1. Introduction

Industrial agriculture has been more widespread recently such as exploiting specialized machinery in the field space, which greatly lightens the farmers’ labor intensity. Because of the application of the Internet of things (IoT) technology and agricultural intelligence, the agricultural machinery becomes a part of the Internet resources. Agricultural machinery scheduling is developing towards the scheduling mode based on an Internet of Vehicles, which causes the gradually increasing demand for agricultural machinery. The scheduling system is crucial to service efficiency. Therefore, how to obtain the most appropriate agricultural machinery (i.e., low cost and high efficiency) with the minimum cost and the shortest time has become the key research direction of agricultural machinery scheduling methods [1]. In addition, it is not practical for every farmer to buy expensive large machinery, so they may have a need for temporary rental machinery. At the same time, the scheduling system also needs to solve the problem of resource waste caused by the idle machinery.

Traditional agricultural machinery scheduling systems are centralized or based on a specific trusted third party [2], where all agricultural machinery owners (AO) are managed by a central scheduling system, and in different areas, establishing a number of central warehouses to place agricultural machinery. In the scheduling process, the center scheduling system receives, processes and stores all data, and applies scheduling algorithm to match AO and agricultural machinery users (AU). That is, every message and datum in the system needs to go through the scheduling system. Current research only focuses on algorithm optimization to improve scheduling efficiency, rather than change the centralized architecture [3]. However, a centralized scheduling system will have the following problems:Single point of failure: the entire system depends on one server, if the server crashes, the entire system will be paralyzed.Low transparency: the data is sent to the user by the central server after passing through the central server. The transparency of scheduling data is relatively low and has retardation. Both sides of the scheduling cannot obtain the data in time, which reduces the decision-making efficiency in the scheduling.High cost: the scheduling system, as a third party, grasps the right of resource dispatching and collects transaction fees from the parties, which directly increases scheduling transaction costs. Centralized system management costs are high, and the cost is eventually borne by the user. Transaction fees are transferred by the bank, which takes a long time and has handling fees.Resource waste: in society, many agricultural machinery owners (AO) possess their own machinery, which can only be idle when not used. Moreover, the machinery managed by the central scheduling system can only be stored in the warehouse when it is idle. These factors caused the low utilization of agricultural machinery, but also the waste of resources.

It is a fact that the agriculture machinery will consume a great number of fossil fuels and induce much environment pollution and soil degradation. As an important means for digital farming the newly-emerging IT technologies take a great role to improve agriculture production efficiency, including blockchain, big data, artificial intelligence and so on. Unlike centralized systems, blockchain technology is decentralized and distributed. Nakamoto first proposed the blockchain concept in 2008 [4]. This technology eliminates the dependence on the third party, makes the transmission and transaction costs extremely low, and improves the transparency of the process [5,6]. Moreover, the knowledge and technology of cryptography guarantee the non-tampering and security of the blockchain [7,8]. In contemporary society, the blockchain technology has been applied in many fields, including supply chain, education, financial industry, etc. [9]. Comparing with the problems of the traditional scheduling method, the advantages of blockchain are as follows:(a)The blochchain is independent of the third party, and the system is maintained by all nodes, so there is no single point of failure.(b)The transaction process data is recorded in the chain, and each user can check it in the chain in time, which improves the transparency and decision-making efficiency.(c)The absence of third parties also means there is no service fee to pay, and transactions are based on digital currency, which is efficient and free of fees.(d)The system based on blockchain can be regarded as a crowdsourcing system, so the idle resources in the society can be fully utilized.

In this paper, a consortium blockchain-based agricultural machinery scheduling system is proposed, aiming to combine the advantages of the blockchain technology, for the intelligent distributed scheduling of agricultural machinery, to obtain the effect of optimal scheduling efficiency and cost. The main contributions are as follows:The system structure has no central platform and adopts the crowdsourced scheduling mode to solve the problems and limitations existing in the traditional agricultural machinery scheduling.Using trusted supervision agencies to verify new block, instead of all nodes validating together. The verifying approach improves the block consensus performance and system security, while solving the supervision problem of consortium blockchain.Taking the consortium blockchain as the system infrastructure, applying the optimized genetic algorithm for scheduling and combining with the supervision, the structure and method of agricultural machinery dispatching are innovated.

## 2. Related Work

The literature [10,11,12] shows some applications of Internet of things technology (IoT) in the agriculture area. Shi et al. [10] indicates that the Internet of Things technology will have a broad prospect in the field of agriculture. Drenjanac et al. [11] and Farooq et al. [12] state that information and data such as the position and status of farmland and agricultural machinery are also more convenient to obtain because of the implementation of IoT in the Agriculture. In a scheduling system, these data are crucial. The algorithm can optimize the scheduling process after processing the data. At present, the following scholars have proposed some optimization algorithms about scheduling. The genetic algorithm is the most used in scheduling.

Ge et al. [13] proposes cloud computing task scheduling based on genetic algorithm, which can effectively schedule computing resources, so that all tasks can be completed in the least time and cost. Jiang et al. [14] proposes a multi-objective, multi-constraint and improved genetic algorithm-based scheduling (MMIGAS) to address the issue of inefficiency of task scheduling. The literature [15,16] proposes a new optimal scheduling algorithm in agricultural machinery scheduling. Jiang et al. [15] proposes the improved fuzzy hybrid immune algorithm which is similar to genetic algorithm to allocate and dispatch from different agricultural machinery resource centers, taking into account the weather, location, time and other situations. It does have a positive impact on scheduling efficiency and transaction cost. The purpose of the algorithm in [2] is to minimize the total travel of the machine and the total execution time of the task. The algorithm, called multi-population co-evolution non-dominant neighbor genetic algorithm (MCNNIA), uses the parameters of geographic data and demand data to optimize matching and scheduling. 

In other aspects, path planning also plays a key role in scheduling, and genetic algorithm is very effective for optimizing path planning. Li et al. [16] proposes an improved genetic algorithm based on a path network to avoid the search cycle. The combination of genetic algorithm and path network can quickly plan the path. By using the path unit, multiple paths can be generated simultaneously, which improves the planning efficiency. Moreover, Lamini et al. [17] proposes an improved genetic algorithm in path planning of a mobile robot. They create a new fitness function considering distance, safety and energy, and improve the crossover operator, to find optimal solutions. Additionally, Xin et al. [18] proposes an improved genetic algorithm for surface vehicle path-planning, which has a positive result for balancing the path-length and time-cost.

Therefore, the genetic algorithm is an effective method to optimize scheduling efficiency and path selection. However, existing agricultural machinery scheduling systems still rely on centralized platforms, which can cause the risk of systemic collapse, high cost and low utilization rates. So, as a decentralized technology, blockchain may solve these problems.

According to Yang et al. [19] and Conoscenti et al. [20], blockchain can be divided into three categories: public chain, consortium chain, and private chain. Consortium chain and private chain can be classified as permissioned blockchain. Public or permissionless blockchains are completely decentralized, with no access control, meaning everyone can participate, and its ledger is completely open and transparent. Therefore, it is suitable for the digital currency field, such as Bitcoin or Ether. However, these characteristics pose great challenges to supervision and privacy protection. Although the transaction and account information is encrypted, it is still easy to disclose due to the contact with real society, which means that it is difficult and impossible to use it in the business field, and it is not enough of the IoT. Instead, the consortium chain requires permission, in which the participants are identifiable and known. The blockchain has a wide application prospect in the commercial field. In addition, transactions and data recorded in the blockchain are traceable and cannot be tampered with after the consensus is reached. This prevents participants from tampering with the ledger and can be used as evidence that the transaction took place. However, Mohanta et al. [9] and Kamilaris et al. [21] consider that blockchain technology can cause the problems of privacy, immature technology, and non-understanding of the operator. It also has the problem of too much reliance on Internet of things devices. 

As for the consortium chain technology, it is very important to ensure that the nodes in the system can maintain a uniform state of ledger, so a reliable and available consensus protocol and algorithm is needed. The literature [22,23,24,25,26] proposes different consensus algorithms—Proof of Work (POW) [22], proof of Stake (POS) [23], practical Byzantine fault-tolerant algorithm (PBFT) [24] and its extended algorithm—Tendermint [25], Hashgraph [26]. These protocols and algorithms are fault-tolerant and improve security. However, the performance of these algorithms is limited, and there are performance bottlenecks in the verification stage, which will bring limitations to the system. Androulaki et al. [27] proposes a well-known permissioned blockchain technology, called Hyperledger Fabric, is also based on PBFT. It uses an execute-order-verify architecture with better throughput to improve performance. Nevertheless, it has more autonomy and requires strong anonymity to prevent privacy breaches, which can make supervision difficult, affect performance and limit its development in certain areas.

Overall, there are also some problems in the consortium chain:Performance problems: in the consensus stage of consortium chain, the verification method has limitations, which affect the performance.Immature supervision technology: proper supervision technology is needed to maintain the normal operation of the consortium chain.

Table 1 shows the contributions and deficiencies of some related works.

As can be seen from the current literature, the existing agricultural machinery scheduling system is based on a central platform to call scheduling algorithms, which means that the problems of centralized platforms mentioned in Section 1 still exist. Moreover, there are shortcomings in the performance and security of the verification method in the consensus method of blockchain. Therefore, inspired by two problems mentioned above, this paper proposes a new agricultural machinery scheduling system based on the consortium blockchain, and then combines with the trusted supervisor to improve the efficiency and security of scheduling. 

## 3. Scheduling System Structure

The scheduling system composition this paper proposes is shown in Figure 1. There are four layers in the system, including certification layer, transaction layer, data layer, and blockchain layer. In addition, there are four participants in the system—certification authority (CA), agricultural machinery owner (AO), agricultural machinery user (AU) and system supervisor (SS), and from time to time, the accounting node (AN) would be randomly selected from all AO and AU nodes to be responsible for block production. Table 2 presents the notations used throughout the paper.

In the certification layer, CA provides AO, AU and SS with authorization to enter the system. Meanwhile, the message propagation in the system adopts asymmetric encryption method, CA issues a public key and private key to these participants, and the public key is open in the system, while the private key is not. Furthermore, the system adopts the authentication access mechanism and the public key is generated by the organization name of the participant. 

In the transaction layer, AN is randomly selected from the nodes of AO and AU, and a new election would be conducted after each AN produces Q blocks. When the AU sends a transaction request, AN invokes the smart contract (SC). Then, as SC does not have direct access to external data, a medium (Oracle) is required to enter the data into SC. 

From the data layer, Oracle obtains the data of farmland, agricultural machinery, weather and road, etc., transmitting them to SC. SC uses the scheduling algorithm and function to calculate the optimal match, sending the result to AN. Then, AN produces the new block, which is verified by SS. After passing, the new block would be recorded in the blockchain layer. In addition, SS monitors the behavior of AO and AU in the transaction layer to ensure the security of the system. Figure 2 shows the interactive structure of the system.

Furthermore, since the data of the scheduling process and the real-time data of the agricultural machinery are also recorded in the blockchain, and the data cannot be tampered with, the participants of the transaction can monitor the data in real-time to understand the state of the agricultural machinery, thus, shortening the decision-making time and improving the decision-making efficiency, and ensuring the authenticity of the data. However, in order to protect some trade secrets and privacy, participants can only track the transactions and data they participate in and cannot obtain other transaction information. This framework can be considered as a crowdsourcing scheduling system. For example, in traditional scheduling, farmers are all used as AU, while in this system, when they have idle machines, they can also be used as AO. Therefore, this scheduling role transformation can make full use of idle resources in society and reduce the waste of resources.

## 4. Scheduling Approach

### 4.1. Scheduling Process

Scheduling process can be divided into three steps, including partition, screening, and functions execution. Table 3 shows the example approach of partition. In this paper, the method based on Geohash which is proposed by Gustavo Niemeyer is adopted, to encode the position [28]. Geohash converts the latitude and longitude of two dimensions into a string. If the string is longer, the range is smaller and the position is more accurate. So the approximate distance between the machinery and the field can be determined by comparing the number of bits that Geohash matches.

For example, encoding the positions of AO_1_ and AO_2_, that are shown in Figure 3:

As for the screening stage, in the region, agricultural machinery models were screened that can satisfy the demand of AU. Next is the matching stage, with three functions, and the meaning of parameters in the function are shown in Table 4:(1)$$F=\alpha T_{mij}+\frac{1}{\beta R}+\eta C$$
(2)$$T_{mij}=\frac{D_{i,j}}{v}*X_{m}*Q_{r}*Q_{w}$$
(3)$$C=\min\overset{i}{\underset{i=1}{\sum }}(C_{i}*A_{i})+\overset{m}{\underset{m=1}{\sum }}C_{m}*\overset{i,j}{\underset{i=1,j=1}{\sum }}D_{i,j}$$

Function (1) (*F*) is the matching objective function, the smaller the value of the objective function, the more matching the agricultural machinery. While function (2) (*T_mij_*) represents the approximate time required for a farm machine to go from field *i* to field *j*, in which the speed is affected by the weather, road conditions and other factors. Function (3) (*C*) is the cost objective function, which is defined as activity cost plus transfer cost. The purpose of this function is to minimize the total cost in scheduling. The genetic algorithm is solved by these functions and the matching result is obtained. 

Therefore, the system adds factors of the organization’s credibility, scheduling costs, weather and road conditions during the matching process. Considering these factors can improve the accuracy and efficiency of scheduling matching.

### 4.2. Smart Contract and Optimal Scheduling Algorithm

Smart contract (SC) plays an important role in the system. SC is responsible for implementing the scheduling matching, price calculation and currency allocation in the system. Therefore, it is crucial to code and monitor the contract, as well as to ensure the security of SC and its implementation process. The code of the contract needs to be carefully checked. Before placing the SC in the system, it is necessary to check the correctness and security of the SC and conduct a trial run. After that, the scheduling system based on the consortium blockchain places the SC in an isolated environment, similar to Fabric’s Docker. Additionally, only SS can access this environment to monitor the execution process of SC, nodes of AO and AU are not entitled to access the environment and modify the contract code, which ensures that an attacker cannot tamper with the code or destroy the SC. Moreover, since the SC cannot actively access external data, a medium is needed to connect the SC to external data. The medium in this system is called Oracle. SC sends a request to Oracle to obtain external data, then Oracle obtains data in the data layer, returning data to the SC. The following pseudocode represents the flow through which the matching process is performed within the SC.
**Algorithm 1.** Smart contract execution algorithm./** Pc: Probability of crossover* Pm: Probability of mutation* T: If the fitness function of any individual produced by evolution exceeds T, the evolution process can be terminated*/  1.Begin  2.Initialize Pm, Pc, M, G, T and other parameters.  3.Pop ← T//Randomly generate the first generation population Pop  4.I ← 0//Evolutionary population algebra  5.Input (farm, machinery and other data)  6.Y ← Initialize(P(I))//Initialize the population,encode geographic location and partition  7.Call matching function  8.while (random (0, 1) < Pc)  9.do [m1 n1] ← corss(newPop,Pc)//Perform crossover operation on 2 individuals with Pc 10.While (random (0, 1) < Pm) 11.do [m2 n2] ← mutation(newPop,Pc)//Perform mutation operation on 2 individuals with Pm 12.newPop ← t//fitness function result of new matching result 13.If t < T then 14.I ++ 15.Else 16.out=output()//Output results, matching orders 17.End

The procedure of Algorithm 1 is based on the genetic algorithm. After initializing parameters and generating first population, the required data is inputted. Then, SC will call functions mentioned in Section 4.1 and mentioned later in this chapter. After that, crossover and mutation will be performed with probability of crossover (Pc) and probability of mutation (Pm), respectively. The next step is to compare the fitness value t after each iteration. If the fitness function of any individual produced by evolution exceeds T, the result will be outputted, and the algorithm will end. Figure 4 shows the flowchart represented by the pseudocode.

After the AN invokes the SC and the SC obtains the data from Oracle, the SC initializes the internal algorithm, loads the data, and inputs the real-time data that is in the data layer. Meanwhile, inputting the encoded locations of agricultural machinery and farmlands, as well as the prospective earning of scheduling. 

The system also incorporates other factors to improve and optimize the scheduling process. This paper defines the function of prospective earning, in order to obtain the prospective earning in the scheduling process, the function is as follows:(4)$$U\left(m\right)=E\left[u\left(m\right)\right]=P_{1}u\left(m_{1}\right)+\ldots +P_{n}u\left(m_{n}\right)$$
where *E*[*u*(*m*)] represents the expected income of agricultural machinery *m_i_*, *i* = 1, 2,…,*n*, the *P_i_*, *i* = 1, 2, …, *n* represents the probability and *u*(*m_i_*) is the benefit to a certain AU of using machinery *m_i_* when *m_i_* is matched with the probability *P_i_*. Thus, the prospective earning of scheduling can be calculated in the algorithm.

Furthermore, in this paper, in the transfer and operation of agricultural machinery, the state transfer equation is used to optimize the path strength selection, the equation is as follows:(5)$$F\left(S_{i,j}\right)=G\left(S_{i,j}\right)+H\left(S_{i,j}\right),$$
where, *G* represents the moving cost of agricultural machinery from farmland *i* to *j*, and *H* is the Heuristic value, and it represents the estimated minimum cost of agricultural machinery from farmland *i* to *j* before the scheduling. After continuous iteration, the minimum value of *F* can be found and the result *J* can be taken:(6)$$J=\min F\left(S_{i,j}\right),$$

Then, position coding uses probability PC and PM in Algorithm 1 for crossover and mutation respectively. Taking the above AO_1_ and AO_2_ as examples, the crossover is displayed in the Figure 5.

And mutation is show in Figure 6.

After the result is obtained, if it is the optimal result, the result is returned to AN and the matching party AO; if not, reload the new data until the result is optimal.

Thus, the scheduling algorithm can be considered as an improved genetic algorithm that combines the classical genetic algorithm with the positioning technique and optimized matching functions. Moreover, the algorithm is called autonomously through SC in the system, rather than through a central server as in traditional systems.

### 4.3. Payment Approach

The transaction payment process of this system is based on digital currency, which eliminates the red tape of transferring money between regional banks, and the extra transaction fees charged by the bank. Although this system does not have the Coinbase transaction that Bitcoin and Ethereum do, it uses existing and legal digital currency. The payment processes are shown in Figure 7. First, Alice (AU) sends a transaction request. After acceptation of the transaction, some of Alice’s funds would be temporarily frozen, in order to prevent Alice from sending attack information, such as denial of service attack (DoS). Then, the AN invokes the SC to match the most optimal result—Bob (AO), and then enters the transaction processing phase. Before this phase, some of Bob’s funds would be also frozen to prevent dishonest and other malicious behaviors in the scheduling process. Their funds would be released after the deal closes. Assuming that Bob acted maliciously in the scheduling process, while the remaining participants were honest, Bob’s frozen funds would be deducted, and allocated to Alice and AN. If the scheduling is normal, AN will invoke the SC that calculates the fees that Alice needs to pay and that Bob should charge. Finally, the payment process is complete. In addition, if AN successfully generates the block, some transaction fees can be obtained, which are determined by the number of transactions contained in the block. 

It should be emphasized that AN is randomly selected from all AO and AU nodes in the system, which means that participating organizations can achieve benefits by generating blocks. This approach can potentially reduce the scheduling cost and reduce the risk of node attacks on the system. Additionally, in order to prevent the double-spending attack, each account has its own balance state, and the account state of all organizations constitutes a state Merkle Tree that maintained by all nodes to prevent tampering. A certain AU cannot send another request to roll back its previous transaction, because when it makes a new request, a new transaction will be triggered with no effect on the previous transaction, that is, this AU would incur new fees. 

### 4.4. Block Consensus Approach

Consensus consists of three stages—election stage, production stage, and verification stage. At the election stage, the system selects a node from the candidate set composed of all AO and AU nodes as the accounting node (AN), and each node in the candidate set has the same probability to be selected. Here, to prevent probability inconsistencies, where an organization creates multiple nodes into the candidate set, the system restricts this behavior—only one node per AO or AU can enter the candidate set. The scheme is feasible because the system implements an access mechanism for identity authentication. After producing Q blocks, a new election would be held. Then, turning to the production stage, that is, the stage that AN produces the new block. If a certain AN cannot produce a block within a certain period of time or it generates an illegal block (the appearance of a ‘fork’ also indicates that the block is illegal), a new AN would be selected. The consensus algorithm pseudocode is shown below.
**Algorithm 2.** Consensus algorithm.  1.Init  2.Round = 0  3.Block = 0  4.upon start do StartRound (0)  5.Select AN  6.Repeat  7.Input data (scheduling request)  8.until data is available  9.Create new block 10.SS verifies new block 11.If block is legal Then 12.Output new block, block ++ 13.Else 14.Select new AN 15.If q < Q then 16.Input new data (scheduling request) 17.Else 18.Round ++, StartRound (Round + 1)

The flowchart of Algorithm 2 is shown in Figure 8.

Furthermore, Figure 7 shows the consensus protocol based on asymmetric encryption in message transmission. (In Figure 9, Node_P represents the nodes of all participants in the system, PK represents public key, and SK represents private key).

AU sends a transaction request with a timestamp, the request is encrypted by the AU’s private key, and the entire message is encrypted by the current AN’s public key. Therefore, AN can use its own private key and AU’s public key to decrypt the request. This method prevents other untrusted nodes from sending forged scheduling requests. The AN verifies that the request is valid. Once verified, the request is executed by invoking the smart contract (SC), generating a block containing multiple scheduling data. Each scheduling can be considered as a transaction (*Tx*), that can be expressed as:(7)$$Tx_{x}=H\left(Tx_{x},t\right)$$

This expression represents the value of the transaction *x* and the timestamp of this transaction obtained by hash function.

Moreover, each block consists of a sequence number, a timestamp, the hash value of the previous block, and the hash value of the root of the Merkle Tree composed of many transactions, as shown in Figure 10. The expression is:(8)$$B_{x}=H(\left(B_{x-1}\right),S_{x},T_{x},H\left(root\right))$$

Meanwhile, data from the scheduling process, such as logistics information and agricultural machinery data, would also be packaged into the blockchain to increase transparency and improve decision-making efficiency. Of course, if the data is encrypted, only the participants in the scheduling can access it, and these participants can only access the scheduling information they participate in, and even AN cannot tamper with the data. After that, AN encrypts the produced block with its own private key and SS’s public key, sending it to SS, and then enters the verification stage. After SS decrypts with its own private key and AN’s public key, SS would verify the block. The block is only valid if SS votes ‘Yes’. Thus, if AN generates an illegal block, SS can find it at the time of verification, preventing the block from taking effect, and then, taking measures of payment deduction and credit deduction against the organization where the AN is located. Meanwhile, the validation process does not require the AO and AU nodes to join, which can directly prevent the system damage caused by the participant’s collusion and the low validation efficiency caused by the participant node dropping. Next, if the verified block is valid, SS would send AN the validation result of the block and its own electronic signature. Thus, the block is successfully generated, and eventually, AN broadcasts to all other nodes in the system, along with the validation results for checking.

This consensual approach ensures the importance of SS and the safe operation of the system. In addition, unlike other consensual methods that require network-wide verification, this verification approach can accelerate consensus efficiency, thereby improving system performance. 

### 4.5. Supervision Approach

Supervision plays a crucial role in the consortium blockchain-based system. Firstly, the system adopts the access mechanism of identity authentication, and the public key and private key are distributed to each participant—AO, AU and SS by CA. The public key is public and is generated by the organization name of the participant, while the private key is not public, is granted by CA, and is saved by AO, AU and SS, respectively. Additionally, this paper assumes that SS is absolutely credible, thus, the privacy of each AO and AU is known to the SS, which ensures that SS is able to discover which real-life participants are acting maliciously. Besides this, SS has the right to monitor the scheduling process, including the scheduling requests sent and the scheduling process data, but it does not participate in the scheduling and affects the scheduling efficiency. For example, if a certain AU sends an attack request that could cause the system to crash, SS can detect it and then take punitive measures of deduction and credit deduction.

In consensus phase, SS is only responsible for block validation and does not participate in the production of blocks. In other words, SS is solely responsible for its verification responsibilities in order to ensure the security and legality of the scheduling process without affecting the efficiency of the scheduling process. As a verification node, it has the deciding vote, which can prevent some malicious behavior of AN. For example, if AN is going to create a ‘fork’, SS can detect and vote ‘No’.

Moreover, only SS can monitor the execution process of SC, ensuring the correctness and security of the execution. Therefore, as shown in Figure 11, SS can guarantee the security of the system from user creation, authentication, information and data authentication, and block verification to SC execution monitoring.

### 4.6. Incentive and Punishment Mechanisms

Since this consortium blockchain-based system does not create its own currency like the public chain, and successful block production would not generate coinbase transactions, it is necessary to use other incentives to encourage the AN not to commit malicious acts. First, when AN is selected, part of its funds should be frozen. If there is an attack on the system in the production process, the funds would be deducted. Instead, the funds are unfrozen at the end of the round, that is, after the production of Q blocks is completed. After the successful production of a new block, AN can receive transaction fees for all transactions included in the block, which can increase the production enthusiasm of AN, include more transactions and improve efficiency. This method also reduces the risk of an attack on the system, as AN would be replaced and punished if it ‘messes up’, the loss outweighs the gain.

Secondly, the concept of credit evaluation is introduced into the system. The parameterization of credit evaluation is actually a weight. The credit is added to the matching function as a parameter and included in the SC. Then the honest node is rewarded in a weighted way, which means that when AN produces a block in an honest way and successfully passes the verification, the organization where the node is located would have a greater chance to be allocated to the scheduling order and obtain economic benefits. In other words, the honest behavior in the system can bring benefits to the organization where the node is located. 

Therefore, maintaining integrity in the consortium blockchain-based system can directly bring economic benefits to the organization. On the contrary, if AN has malicious behavior, its organization would be punished by deducting its credit evaluation parameter, which makes it difficult to match the order, and would also be fined, which directly affects the income.

## 5. Result and Discussion

The experimental results show that the proposed agricultural machinery scheduling system can solve the problems of single point crash, high cost, and waste of resources. Meanwhile, we test the availability and effectiveness of the system by simulation.

Firstly, this paper constructs two scenarios—high demand for agricultural machinery and low demand for agricultural machinery, for comparing the traditional centralized scheduling method with the method proposed in this paper. We assume that the daily demand for the machinery is 10,000Q at high demand and 1000Q at low demand. Meanwhile, we assume that other things like scheduling prices are the same; the results are shown in Table 5.

We compare two scheduling systems by testing their security, scheduling cost and utilization rate of machinery. Thus, according to Table 5, When demand is high and low, the traditional method may suffer from the system crash. In contrast, this proposed system is more secure. Additionally, regardless of if the demand is high or low, the scheduling cost of the traditional method is 2000 to 4000 RMB higher than that of the system proposed in this paper. And the utilization rate of agricultural machinery is the same. As shown in Table 5, when demand is high, the utilization rate of the centralized system and that of the blockchain-based system is 82% and 95%, respectively. When demand is low, the utilization rate of the former drops to only 46%, but the latter remained stable at more than 70 percent. Thus, obviously, the scheduling system proposed by this paper has a higher utilization rate of agricultural machinery, which means that it can cause less resource waste.

Then, we create the second case—the different farmland coverage per 200,000 square kilometers of land area. The two scenarios are 10% and 30%, respectively. We assume that other factors like demand and machinery dispatch price are the same. The results are shown in Table 6.

The parameters we compared are the same as in the first scenario, as shown in the table, the results are similar. Compared with the traditional centralized scheduling approach, the method proposed in this paper performs well in security, cost reduction, and machinery utilization rate.

Therefore, Table 5 and Table 6 show that, compared with the centralized system, the consortium blockchain-based scheduling system can effectively increase safety, reduce the scheduling cost, and improve the utilization rate of machinery.

Moreover, to verify the availability, effectiveness, and breakthrough of the proposed method, we conduct the experiments to test the relationship between distance and time in the scheduling process of the system and also test the relationship between the evolution algebra of the objective function and the optimal solution. The experiments adopt Algorithm 1. The experimental results in Figure 12 show the simulation results of the system proposed in this paper. 

Figure 12a shows the optimization variable results to verify whether this scheduling system can meet the requirements of whether the agricultural machinery can arrive at a certain time within a certain distance. The y-coordinate is the distance that agricultural machinery should travel in the scheduling process, and the x-coordinate is the time taken for the agricultural machinery to travel a certain distance. The red line is the demand and the blue line is the real situation by using this system. From Figure 12a, it can be seen that the blue line (real situation) is close to the red line (demand) and sometimes even above the red line, which means by using this system, in most cases, the time used for agricultural machinery scheduling is close to or even shorter than the time required by users. Among them, because the weather and road factors are also considered in the scheduling of the system, in order to be closer to the actual life situation, we add extreme weather and road conditions in the simulation experiment, that is, the position of x-axis close to 10 h and 15 h in the figure, which is also the reason why the blue line is so much below the red line. 

In addition, Figure 12b shows the optimization function result. The experiment is based on Algorithm 1. In the figure, the y-coordinate is target value, and the x-coordinate is evolution algebra. The experiment set the termination algebra to 500. The red curve represents the average value of each generation, and the blue curve represents the best value of each generation. As shown in Figure 12b, the objective function can reach the optimal solution that is the best matching when the algorithm iterates for 400 times.

Finally, the verification performance of the block consensus in this system is verified by simulation. We compare the traditional block validation approach with the improved approach in this article that is based on Algorithm 2, and the results are shown in Figure 13.

In Figure 13, the abscissa represents the validation time, and the ordinate represents the number of blocks to be validated. The red line represents the conventional consensus efficiency that requires all nodes to validate together, and the blue line represents the efficiency of the system supervisor (SS)-based approach. Obviously, the blue line is basically always above the red line, and the improved verification method is much better than the previous method when the time exceeds about 5T.

Therefore, through the experiment, we achieve positive results about the consortium blockchain-based agricultural machinery scheduling system. Firstly, the system performs better than the centralized scheduling system in terms of security, cost and resource utilization. Secondly, the experiment shows that the scheduling method of the system can meet the requirements and the matching equation and Algorithm 1 are optimized. In addition, compared with previous consensus verification method like POW [22] and PBFT [24], the implementation of block verification method based on the system supervisor (SS) improves the consensus performance of the system.

However, the use of the blockchain-based system in agriculture will pose challenges. Kamilaris et al. [21] stated that the blockchain training platforms are scarce, and there is a shortage of understanding between policymakers and technical experts. Zhao et al. [29] illustrated that the performance of blockchain-based systems may be affected when the transaction volume is high. Galvez et al. [5] considered that imperfect policy in the blockchain domain can cause the wrong business decisions. They also consider that the blockchain system still relies on Internet of things technologies like RFID for detection, which cannot fundamentally solve the problem of data fraud. According to these researches, in agricultural machinery scheduling, using consortium blockchain technology may also suffer from the problems of implementation difficulties, performance bottlenecks and problems with falsification of machinery and field data.

But blockchain technology has undeniable advantages. Nowadays, in scheduling fields, blockchain has been used to solve some problems. Zhao et al. [29] study and analyze the microgrid market transaction system based on the consortium blockchain, which can also be regarded as the microgrid scheduling system. The system solves the problem of high cost and poor security. The system is faster, better performance and better security. Articles [30,31] also propose the energy trading system based on blockchain. Aitzhan et al. [30] combine multiple signatures and anonymous encryption message propagation flow technology to protect user privacy and information security, and Son et al. [31] encrypt all bids and matches encrypted bids to each other through a smart contract based on functional encryption. Hîrţan et al. [32] present a reputation system for Intelligent Transportation Systems, that can be considered as blockchain-based vehicle dispatching system. The traffic information and data are jointly verified by users, which ensures data reliability and avoids traffic congestion.

On the other hand, in the traceability system of the supply chain, Venkatesh et al. [33] propose a blockchain-based supply chain system, which is combined with Internet of Things and Big Data technology. The aim is to increase the transparency of the supply chain to monitor it and effectively increase social sustainability. George et al. [34] propose a restaurant prototype by using blockchain to improve food traceability. And it combines with food quality index algorithm which can determine the health of the food, to satisfy traceability and health requirement in society. In addition, in terms of cost control, Schmidt et al. [35] state that the blockchain effectively reduces the transaction cost in the supply chain, and it also improves transaction transparency.

Obviously, the blockchain-based system performs well in security, cost control, traceability and transparency, which is consistent with the results in this paper. Besides this, the system in this paper also improves resource utilization and block validation performance. However, this technology still has challenges in imperfect policy, falsification of data and expansibility, which causes implementation difficulties. For the future works, we will carry out further research and improvement on these problems.

## 6. Conclusions

This paper presents an intelligent agricultural machinery scheduling system based on consortium blockchain. Unlike some well-known blockchains, it is a non-monetary blockchain. In addition, unlike the traditional center-based scheduling method, this system does not need a central server, but relies on some smart contracts in the system to execute the scheduling and all nodes together to maintain the ledger. The system addresses some of the problems associated with traditional centralized systems, and solves the problem of slow validation efficiency in traditional block consensus:(1)Reduce transaction costs: there is no centralized platform in this scheduling system to charge fees. Although the participants need to pay a small transaction fee to AN, the nodes would have the opportunity to become bookkeeping nodes, thus bringing benefits to the organization. In addition, the system’s transactions are based on digital currency. In this way, there is no need to pay the bank transfer fees, but also payment time is reduced.(2)It does not depend on the center, which means there is no central server controlling system, which can prevent a single point of the system crash.(3)The scheduling system based on the consortium blockchain can also make full use of the idle resources in the society. In this system, AUs cannot only have a demand for agricultural machinery, but also can provide their own agricultural machinery that they do not need or do not use for others to use.(4)In the matching function and algorithm of scheduling, the system combines the classical genetic algorithm and considers the factors of weather, road, cost, benefit and organizational reputation. The experimental results show that it has an optimization effect on scheduling.(5)Due to the advantages of blockchain, the transaction process and data can be tracked, and also can be acquired in real-time, making the system auditable and traceable, and improving the decision-making efficiency.(6)In the process of block consensus, the verification block is verified by the trusted supervisor (SS) rather than by all nodes in the system, which improves the verification efficiency and improves the performance of the system.(7)Ensure security through penalties and incentives, supervision and isolation of SC. The system also guarantees that nodes unrelated to a transaction cannot obtain any information about the transaction to protect business privacy. Furthermore, the use of cryptography makes it impossible for data in the system to be tampered with. Therefore, the consortium blockchain-based agricultural machinery scheduling system has many improvements and advantages compared with the previous scheduling system. Meanwhile, SS can also monitor the user behavior in the system, monitoring the implementation of the smart contract (SC) process, so that the security of the system is improved.

Nevertheless, the simulation scenario of the system is relatively simple. There are still some challenges in policy improvement, system scalability and data verification. Thus, future work is needed to verify whether the system can work well in practical application scenarios.

## Figures and Tables

**Figure 1 sensors-20-02643-f001:**
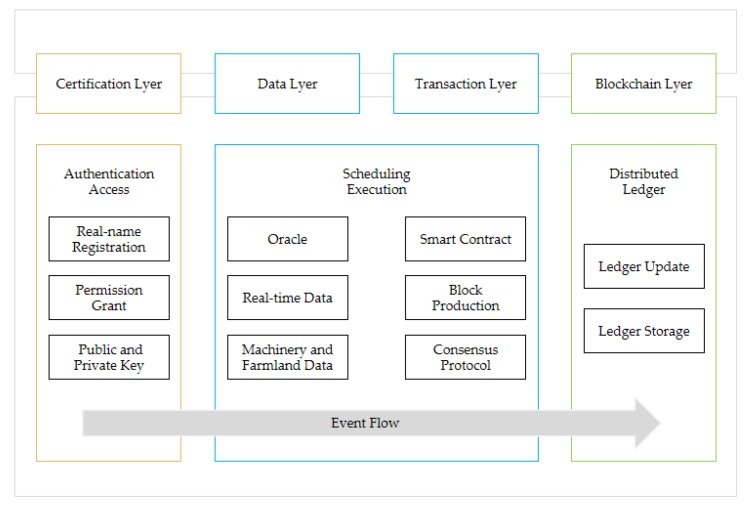
System composition.

**Figure 2 sensors-20-02643-f002:**
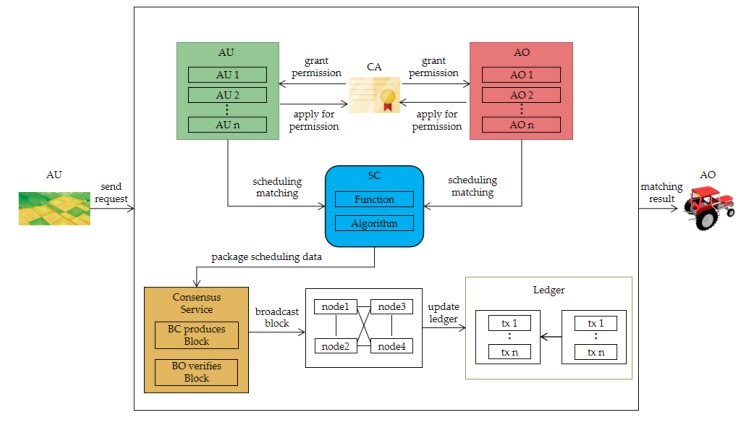
Interactive structure of the system.

**Figure 3 sensors-20-02643-f003:**
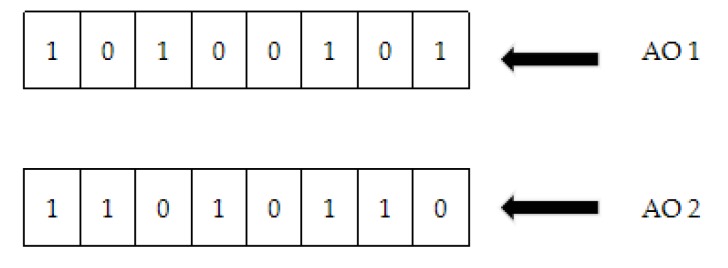
Position encoding.

**Figure 4 sensors-20-02643-f004:**
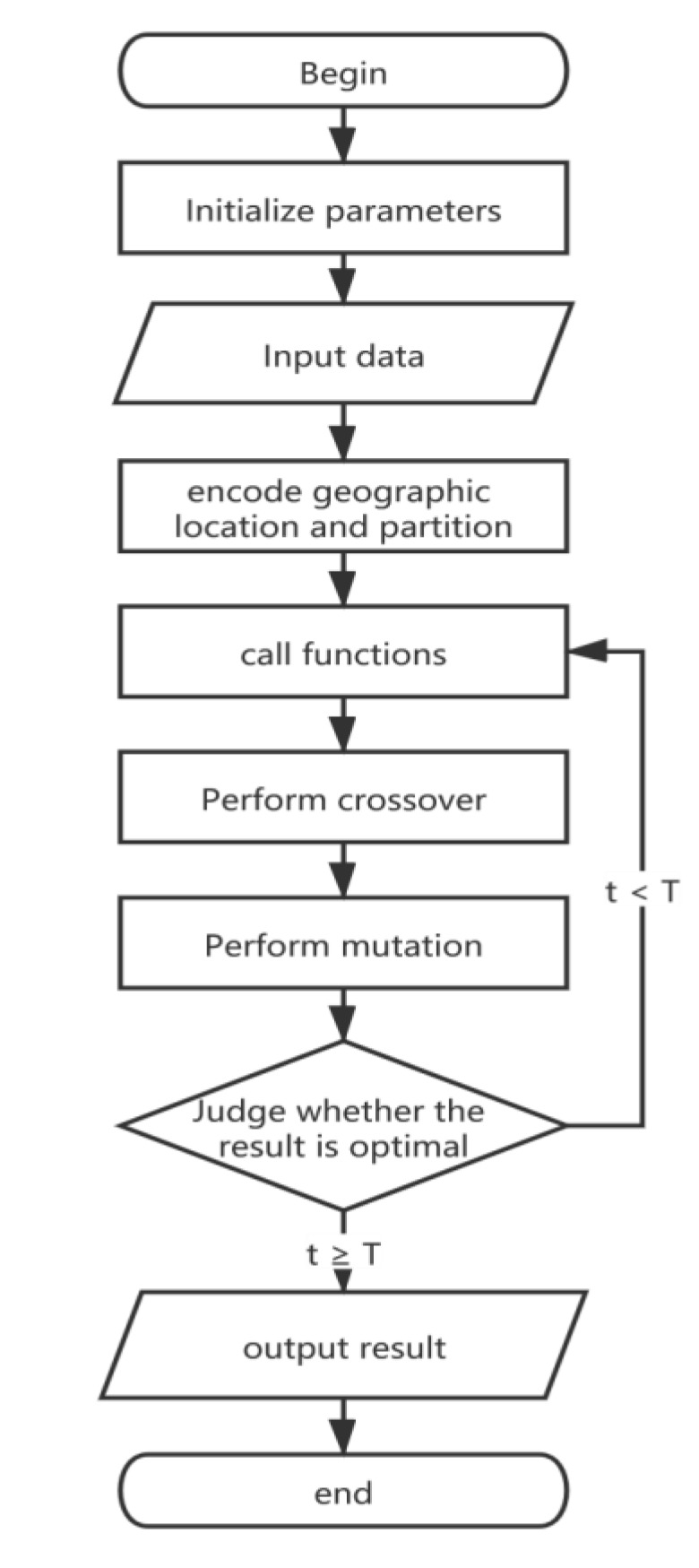
Algorithm 1 Flowchart.

**Figure 5 sensors-20-02643-f005:**
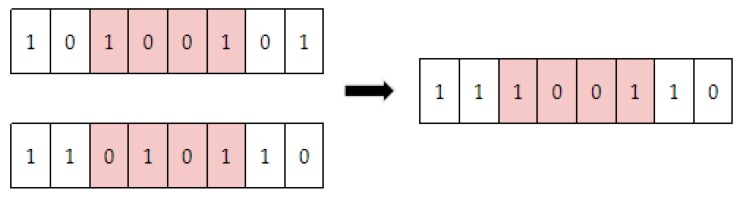
Crossover.

**Figure 6 sensors-20-02643-f006:**
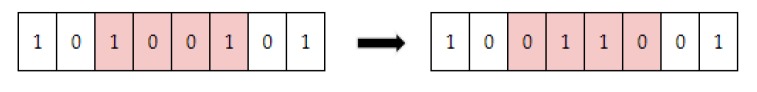
Mutation.

**Figure 7 sensors-20-02643-f007:**
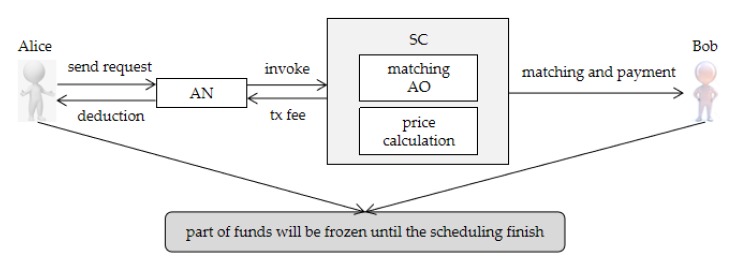
Payment process.

**Figure 8 sensors-20-02643-f008:**
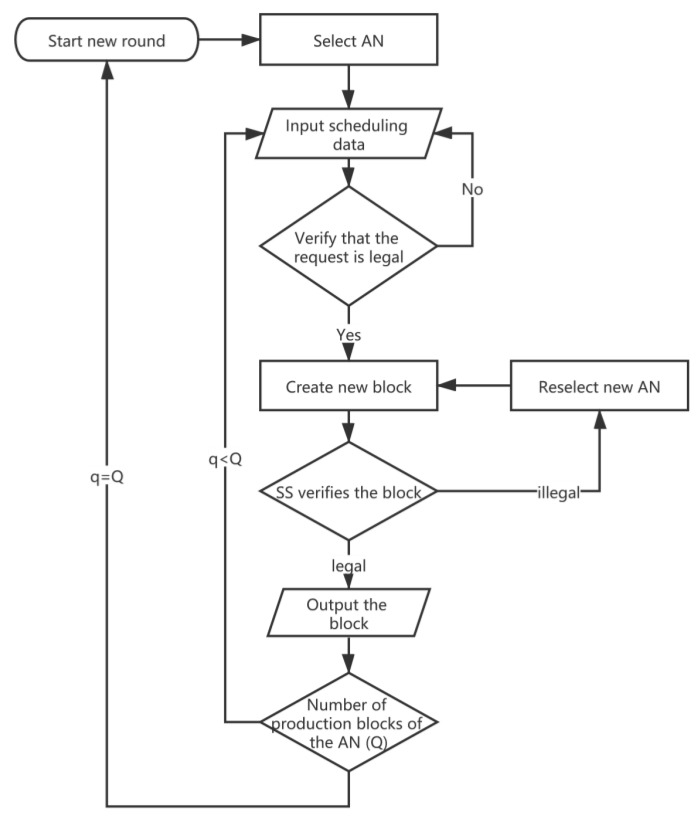
Algorithm 2 Flowchart.

**Figure 9 sensors-20-02643-f009:**
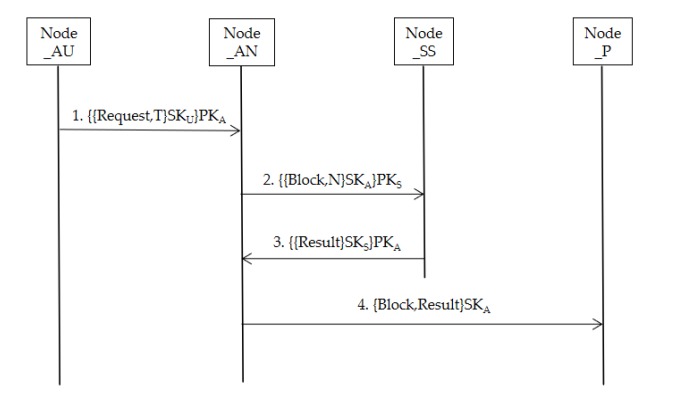
Consensus protocol.

**Figure 10 sensors-20-02643-f010:**
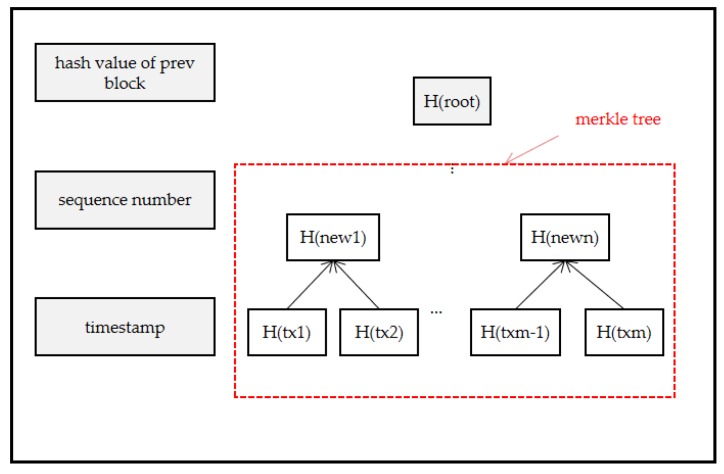
Block structure.

**Figure 11 sensors-20-02643-f011:**
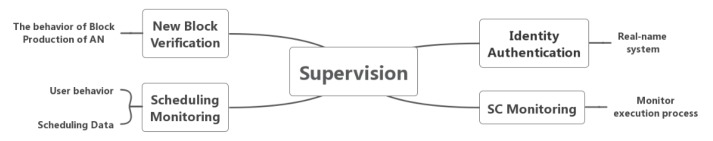
Supervision in system.

**Figure 12 sensors-20-02643-f012:**
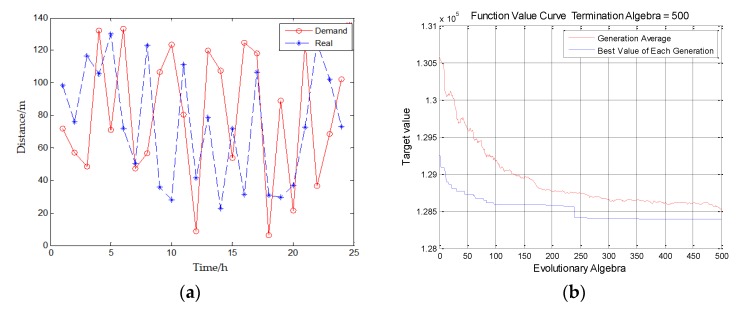
Experimental results: (**a**) optimization variable result; (**b**) optimization function result.

**Figure 13 sensors-20-02643-f013:**
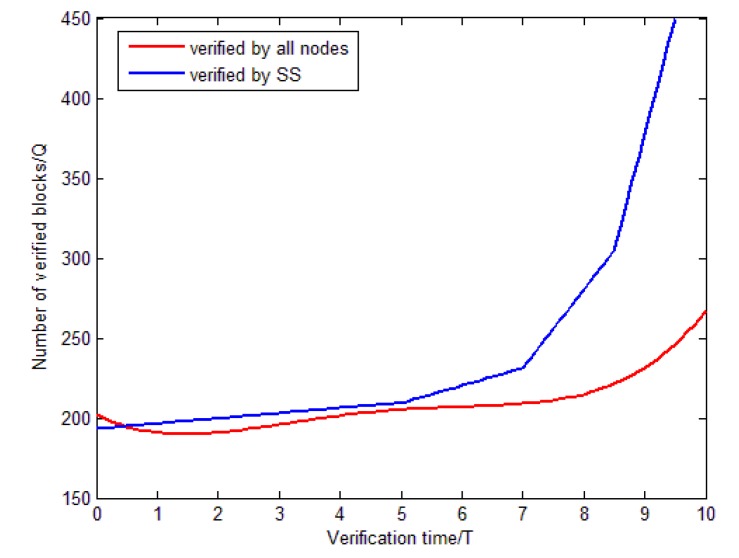
Verification efficiency comparison.

**Table 1 sensors-20-02643-t001:** Contributions and Deficiencies of Related Works.

Literature	Contributions	Deficiencies
[10,11,12]	Expound the importance of Internet of things technology in the field of agriculture, as well as its future development and related technology description	____
[2,13,14,15]	Based on the centralized scheduling method, the scheduling algorithms: genetic algorithm is optimized to improve the scheduling efficiency	These cannot solve the problem of centralization—single point of failure, low transparency, high cost and waste of resources
[16,17,18]	Genetic algorithm is proposed to improve the precision and path-planning, as well as the scheduling efficiency	____
[9,19,20,21]	Review of research on applications in different fields, especially IoT, and analyze the advantages and disadvantages of different types of blockchain	Blockchain may cause privacy issues, few training platforms, immaturity problems, and cannot solve the problem of data fraud
[22,23,24,25,26]	Blockchain consensus protocol and algorithm: POW, POS, Byzantine fault-tolerant algorithm. Improve system fault tolerance.	Performance, validation efficiency and availability are insufficient
[27]	New consortium chain architecture, execute-order-verify architecture, improves system performance; use of membership mechanism to limit the access permission	The lack of supervision technology

**Table 2 sensors-20-02643-t002:** Notations.

Symbols	Meaning
CA	Certification Authority
AO	Agricultural machinery Owner
AU	Agricultural machinery User
SS	System Supervisor
AN	Accounting Node
SC	Smart Contract

**Table 3 sensors-20-02643-t003:** Geohash [28].

Geohash Length	Lat Bits	Lng Bits	Lat Error/Degrees (°)	Lng Error/Degrees (°)	Km Error/km
1	2	3	±23	±23	±2500
2	5	5	±2.8	±5.6	±630
3	7	8	±0.70	±0.70	±78
4	10	10	±0.087	±0.18	±20
5	12	13	±0.022	±0.022	±2.4
6	15	15	±0.0027	±0.0055	±0.61

**Table 4 sensors-20-02643-t004:** Scheduling parameters.

Parameter	Meaning
$T_{mij}/h$	Time required of machinery *m* to get from field *i* to *j*
R	Credit
C/RMB	Scheduling cost
$D_{i,j}/m$	The distance between field *i* and *j*
v/m/h	Average speed of machinery
$X_{m}$	When the machine is idle, the value is 1; when the machine is working, the value is 0
α,β,η	Weight coefficient
$Q_{w}$	Greater than or equal to 1, the better the climate, the closer to 1
$Q_{r}$	greater than or equal to 1, the better the road condition, the closer to 1
$\overset{i}{\underset{i=1}{\sum }}C_{i}/\mathrm{RMB}$	Operation cost per unit area per farm
$A_{i}{/m}^{2}$	The area of each farm
$\overset{m}{\underset{m=1}{\sum }}C_{m}/\mathrm{RMB}$	Transfer cost per unit distance of farm machinery
$\overset{i,j}{\underset{i=1,j=1}{\sum }}D_{i,j}/m$	The sum of the distances between each field

**Table 5 sensors-20-02643-t005:** Demand based scenario comparison.

	High Demand (10,000Q)			Low Demand (1000Q)		
	Security	Cost	Utilization	Security	Cost	Utilization
**Centralized method**	Potential paralysis	17,274.88	82%	Potential paralysis	12,479.21	46%
**Method of this paper**	High	13,050.01	95%	High	10,215.47	74%

**Table 6 sensors-20-02643-t006:** Scenario comparison based on farmland occupation.

	Low Share of Farmland (10%)			High Share of Farmland (30%)		
	Security	Cost	Utilization	Security	Cost	Utilization
**Centralized method**	Potential paralysis	10,258.31	42%	Potential paralysis	18,007.33	78%
**Method of this paper**	High	8478.07	74%	High	13,358.47	93%

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
