# Peer review of "A Consortium Blockchain-Based Agricultural Machinery Scheduling System"

_sensors, 2020, doi:10.3390/s20092643_

Round 1

Reviewer 1 Report

In this paper, a consortium blockchain-based agricultural machinery scheduling system is proposed, in order to solve the problems of the traditional centralized scheduling system, such as single point of failure, high cost, low transparency, waste of resources and so on. This study will be very interesting for many readers, however, the followings should be corrected.

(1) As the explanations of Result in Chapter 5 are insufficient, the superiority (effects) of the proposed method cannot be sufficiently understood. Additional information should be required as follows.

・In table 4, the proposed method was compared with the traditional method about 2 scenario of high demand and low demand for agricultural machinery. However, since there is only one case study, the superiority of the proposed method cannot be sufficiently verified.

・The meaning of high demand and low demand for agricultural machinery (line 408~409) is not clarified. Moreover, the traditional scheduling method is not clarified.

・With no sufficient explanation of Figure 10, it is very difficult to understand. Therefore, the effects of the proposed method cannot be confirmed.

(2) The explanation of Section 4.2 (Smart contract and optimal scheduling algorithm) is difficult to understand.

・line 264~264:“the probability Pi is mi, i=1,2… N, and the utility of an AU is u(mi) when it definitely gets mi.”is incomprehensive.

・line 269:”the estimated cost” is incomprehensive.

Author Response

Dear reviewer,

Sincerest thanks for your kind encouragement and constructive comments on our manuscript titled "A Consortium Blockchain-Based Agricultural Machinery Scheduling System" (paper ID: sensors-774760). We have thoroughly revised the paper in response to the insightful comments. We hope that all these modifications can meet your remarks. In the following, we will respond to detailed comments one by one. For easy reference, our replies to each of comments are highlighted in bold and blue fonts.

************************************

‘In this paper, a consortium blockchain-based agricultural machinery scheduling system is proposed, in order to solve the problems of the traditional centralized scheduling system, such as single point of failure, high cost, low transparency, waste of resources and so on. This study will be very interesting for many readers, however, the followings should be corrected.’

Reply:

We thank you for your kind and insightful comments. Let us explain as follows. In the following, each of your comments is replied accordingly.

Comment 1:

(1) As the explanations of Result in Chapter 5 are insufficient, the superiority (effects) of the proposed method cannot be sufficiently understood. Additional information should be required as follows.

In table 4, the proposed method was compared with the traditional method about 2 scenario of high demand and low demand for agricultural machinery. However, since there is only one case study, the superiority of the proposed method cannot be sufficiently verified.

Reply:

We thank you for your kind advice. We have carefully checked and corrected the whole manuscript as you suggested. The previous Table 4 is now changed to Table 5 because we added Table 2 above -- Notations. Then we added Table 6 to create another case study -- the different farmland coverage per 200,000 square kilometers of land area. The two scenarios are 10% and 30%, respectively.

Comment 2:

The meaning of high demand and low demand for agricultural machinery (line 408~409) is not clarified. Moreover, the traditional scheduling method is not clarified.

Reply:

For the meaning of high and low demand for the machinery and traditional centralized scheduling method, now we have made modifications and explanations in the paper (line 455-457 ) and in the table. High and low demand -- the daily demand for the machinery is 10000Q at high demand and 1000Q at low demand. Meanwhile, we assume that other things like scheduling prices are the same. And we changed the word ‘traditional’ to ‘centralized’ in Table 5.

Comment 3:

With no sufficient explanation of Figure 10, it is very difficult to understand. Therefore, the effects of the proposed method cannot be confirmed.

Reply:

The previous Figure 10 is now changed to Figure 12 because we added Figure 4 and Figure 8 above. And we have expanded the discussion and explanation of this result -- The definition of the axes, the definition of the lines and the explanation of the results are illustrated. We also conduct another experiment for testing the performance of consensus, and the result Figure 13 is added into the result section.

Comment 4:

(2) The explanation of Section 4.2 (Smart contract and optimal scheduling algorithm) is difficult to understand.

・line 264~264:“the probability Pi is mi, i=1,2… N, and the utility of an AU is u(mi) when it definitely gets mi.”is incomprehensive.

・line 269:”the estimated cost” is incomprehensive.

Reply:

Now in line 291-293. We apologize that it is really lack of explanation. We change the interpretation of equation (4) to: mi, i = 1, 2,...,n, the Pi, i = 1, 2,...,n represent the probability. and u(mi) is the benefit that an AU of using machinery mi when mi is matched with the probability Pi.

Line 297-299. And we change ‘the estimated cost’ to ‘the Heuristic value’ which represents the estimated minimum cost of agricultural machinery from farmland i to j before scheduling starts.

……………………………………………

Thank you for your consideration of our manuscript.

Yours sincerely,

Dr. Xiong.

Reviewer 2 Report

The authors carried out a study on the structure of an agricultural machinery scheduling system using blockchain technology helping to maximize a community's use of agricultural machinery. The main idea is original and interesting, but the design of the manuscript should be improved as well as the result section before publishing.

It is described well the problems that other systems have and the advantages of blockchain, justifying the purpose of the work. The paper has a good description of the methods but has not a good results section. Moreover, the manuscript requires a new discussion section too, compulsory for the quality of this journal. General comments are:

  • You should change the keywords with different words to the title to enhance the search on databases.
  • Line 39: “and drones in the field space,”. It has no sense.
  • Line 56: “Current 56 research…”. Add references.
  • Line 104. “These papers demonstrate that IoT makes agriculture more intelligent”. Please, change this sentence.
  • Line 108. “At present, some scholars have proposed some optimization algorithms about 109 scheduling.” Who?
  • It is advisable to check the position of the references in the sentences, showed as subject of the sentences. That does not sound well.
  • Table 1 shoud be merged into the text either in the discussion section or in the related work section. The table not distinguish between contribution and deficiencies.
  • In my opinion, it is not necessary to subhead section 2.1,2.2, 2.3, 2.4, 2.5.
  • After line 182, seems that material and methods section starts, so delimit this section. Before that, it is necessary to summarize the main objective of this work because is Line 179-181 are very confused.
  • Please, provide a list of abbreviations that are very extensive in this work. 
  • Line 223. What is Geohash? At least a brief description is necessary
  • Table 2. Add the error units.
  • Table 3. should be located after the equations.
  • Table 3. Which are the units of the scheduling parameters?
  • Line 257. Pseudocode is not  referenced  in Figure or Table.
  • Algorithm 1 pseudocode. Point 10 is in another font/size
  • Figure 6. Review text in the figure: “requedt” will be “request”
  • Line 323. Pseudocode is not  referenced  in Figure or Table.
  • Figure 7. Review text in the figure: “Repuest” and “Rsult” wil be “Request” and “Result”. In addition, Node_BC, Node_BO and Node_P are not defined in the text.
  • Line 307, 338. Change “Merkel Tree” by “Merkle Tree”
  • Line 322. Algorithm 2 pseudocode.: “1. Init” is in bold
  • Line 322. Algorithm 2 pseudocode. Acronyms “BO” and “BC” are not defined in the text
  • Equations in page 11, line 334, 339, are not numbered
  • Table 4. Tradition by Traditional.
  • Results section require a further explanation of the method used for validate problem statement, the simulation performed of the parameters that are calculated. Please,
  • Result section requires be extended and a more in-depth discussion as well as the provision of references.
  • In this regard, some of the data set out in section 2, could be relocated in the discussion of the work after the results obtained, but also other things should be added.
  • References section has a huge quantity of conferences. This may harm the quality indicators of the journal. Perhaps, if the manuscript changes its composition it could be enhanced, reducing the conferences citation and enlarging the publication in scientific journals. I would suggest to discuss the implication of the proposed system in other issues such as traceability system, connections with machinery, new challenges,  and others you consider, where you can add these references proposed:

https://doi.org/10.1016/j.tifs.2019.07.034

https://doi.org/10.1016/j.compag.2019.104980

https://doi.org/10.1016/j.trac.2018.08.011

10.1109/ACCESS.2018.2890507

  • Conclusion report the advantages of the work approach but it is necessary to mention the main conclusion extracted from the missing section “discussion” that may content a critical discussion of the weaknesses of this work in the face of the challenges posed, or a comparison with other works.

Author Response

Dear reviewer,

Sincerest thanks for your kind encouragement and constructive comments on our manuscript titled "A Consortium Blockchain-Based Agricultural Machinery Scheduling System" (paper ID: sensors-774760). We have thoroughly revised the paper in response to the insightful comments. We hope that all these modifications can meet your remarks. In the following, we will respond to detailed comments one by one. For easy reference, our replies to each of comments are highlighted in bold and blue fonts.

************************************

‘The authors carried out a study on the structure of an agricultural machinery scheduling system using blockchain technology helping to maximize a community's use of agricultural machinery. The main idea is original and interesting, but the design of the manuscript should be improved as well as the result section before publishing’.

Reply:

We thank you for your kind and insightful comments. In the following, each of your comments is replied accordingly.

Comment 1:

It is described well the problems that other systems have and the advantages of blockchain, justifying the purpose of the work. The paper has a good description of the methods but has not a good results section. Moreover, the manuscript requires a new discussion section too, compulsory for the quality of this journal. General comments are:

‘You should change the keywords with different words to the title to enhance the search on databases.’

Reply:

We have deleted the Keyword ‘Agricultural machinery scheduling’, and added ‘Matching function; Scheduling algorithm; Consensus algorithm’. Also, we have changed the abstract accordingly.

Comment 2:

‘Line 39: “and drones in the field space,”. It has no sense..’

Reply:

We have deleted ‘and drones’ on page 1, line 40.

Comment 3:

‘Line 56: “Current 56 research…”. Add references..’

Reply:

We have added the reference [3] on page2, line 58.  

Bochtis, D. D.; Sørensen, C. G. C.; Busato, p. Advances in agricultural machinery management: A review. Biosystems Engineering 2014, 126, 69-81.

Comment 4:

‘Line 104. “These papers demonstrate that IoT makes agriculture more intelligent”. Please, change this sentence.’

Reply:

Page 3, line 108-112. We have deleted this sentence, and changed to ‘Shi et al. [10] indicates that the Internet of Things technology will have a broad prospect in the field of agriculture. Drenjanac et al. [11] and Farooq et al. [12] state that information and data such as the position and status of farmland and agricultural machinery are also more convenient to obtain because of the implementation of IoT in the Agriculture.’ We have introduced each of the literature to help understand it better.

Comment 5:

‘Line 108. “At present, some scholars have proposed some optimization algorithms about 109 scheduling.” Who?’

Reply:

Line 113. We changed ‘some scholars’ to ‘the following scholars’

Comment 6:

‘It is advisable to check the position of the references in the sentences, showed as subject of the sentences. That does not sound well.’

Reply:

In Related Work section, we have changed the position of references in the sentences. We changed each sentence in which the subject is the references to the author’s name + [reference]. In addition, we changed the description of reference [15] ([13] in the original version), and changed the position of [14] (original [15]) in the text, which makes the description of literature more accurate.

Comment 7:

‘Table 1 shoud be merged into the text either in the discussion section or in the related work section. The table not distinguish between contribution and deficiencies..’

Reply:

We placed Table 1 in the Related Work section. And Table 1 has been distinguished between contribution and deficiencies. We also added description of deficiencies and challenges aspects of blockchain literature -- [9], [19-21].

Comment 8:

‘it is not necessary to subhead section 2.1,2.2, 2.3, 2.4, 2.5.’

Reply:

We have deleted these subheads.

Comment 9:

‘after line 182, seems that material and methods section starts, so delimit this section. Before that, it is necessary to summarize the main objective of this work because is Line 179-181 are very confused..’

Reply:

We have changed those sentences. Now on page 5, line 177-183, firstly, we summarize the deficiencies in related work -- ‘It can be seen from the current literature, the existing agricultural machinery scheduling system is based on the central platform to call scheduling algorithm, which causes the fact that the problems of centralized platform mentioned in section 1 still exist. Moreover, there are shortcomings in the performance and security of the verification method in the consensus method of blockchain.’ Then, we summarize the main objective of our work -- ‘Therefore, inspired by two problems mentioned above, this paper proposes a new agricultural machinery scheduling system based on the consortium blockchain, and then combines with the trusted supervisor to improve the efficiency and security of scheduling’.

Comment 10:

‘Please, provide a list of abbreviations that are very extensive in this work.’

Reply:

We have added Table 2. Notations on the page 6.

Comment 11:

‘Line 223. What is Geohash? At least a brief description is necessary

Table 2. Add the error units.’

Reply:

On page 7, line 233-234, we added a brief description about Geohash, and added a reference [28] that is a web.

Geohash.org. Available online: http://geohash.org/site/tips.html

The previous Table 2 has become Table 3. And unit has been added -- the Lat error/degrees (°), Lng error/degrees (°), KM error/km.

Comment 12:

‘Table 3. should be located after the equations.

Table 3. Which are the units of the scheduling parameters?.’

Reply:

The previous Table 3 has become Table 4. we have placed Table 4 after the equations on page 8.

And we have added the units of parameters into Table 4.

Comment 13:

‘Line 257. Pseudocode is not referenced in Figure or Table.’

‘Line 323. Pseudocode is not referenced in Figure or Table.’

Reply:

We added Figure 4. Algorithm 1 Flowchart on page 10 and Figure 8. Algorithm 2 Flowchart on page 13 to make Pseudocode easier to understand. Also, in result section, we illustrated the Figure 12 and Figure 13 are based on Algorithm 1 and Algorithm 2 respectively.

Comment 14:

‘Algorithm 1 pseudocode. Point 10 is in another font/size.’

Reply:

Line 272. In Algorithm 1, we changed the Point 10 to the normal size. Moreover, we improved algorithm 1 to reflect genetic algorithm more, and added annotation to the algorithm.

Comment 15:

‘Figure 6. Review text in the figure: “requedt” will be “request”.’ ‘Figure 7. Review text in the figure: “Repuest” and “Rsult” wil be “Request” and “Result”. In addition, Node_BC, Node_BO and Node_P are not defined in the text.

Line 307, 338. Change “Merkel Tree” by “Merkle Tree”’

Reply:

We apologize for such a careless mistake. Spelling errors have been corrected in Figure 7 (original 6) on page 11 and Figure 9 (original 7) on page 13, and in Figure 9 (original 7), BO changed to SS, BC to AN, and Node_P explained.

We have changed ‘Merkel’ to ‘Merkle’ in line 341, 378 and Figure 10 (original 8).

Comment 16:

‘Line 322. Algorithm 2 pseudocode.: “1. Init” is in bold

Line 322. Algorithm 2 pseudocode. Acronyms “BO” and “BC” are not defined in the text.’

Reply:

Line 356. We have changed ‘1. Init’ to non-bold. And changed ‘BO’ to ‘SS’, ‘BC’ to ‘AN’ in Algorithm 2.

Comment 17:

‘Equations in page 11, line 334, 339, are not numbered’

Reply:

We have numbered these two equations (7) on page 13, line 374 and (8) on page 14, line 379.

Comment 18:

‘Table 4. Tradition by Traditional..’ Results section require a further explanation of the method used for validate problem statement, the simulation performed of the parameters that are calculated. Please,

Reply:

The previous Table 4 has become Table 5. We have changed ‘Traditional’ to ‘Centralized’, in order to facilitate the understanding.

Comment 19:

‘Results section require a further explanation of the method used for validate problem statement, the simulation performed of the parameters that are calculated. Please,’

Reply:

We have conducted anther simulation experiment about block verification. The result is shown in Figure 13 (page 17). And we also provide the explanation to the result.

Comment 20:

‘Result section requires be extended and a more in-depth discussion as well as the provision of references.

In this regard, some of the data set out in section 2, could be relocated in the discussion of the work after the results obtained, but also other things should be added.’

Reply:

We change the title of Section 5 to ‘Result and Discussion’, and we added another case study which is shown in Table 6. We have expanded the discussion and explanation of this result. We describe the parameters and results in the figures and tables. After line 514 on page 17, it’s Discussion section.

In addition, we took the advice, the original part about the application of blockchain in section 2 (Original line 148-158) is put in the discussion section, line 531-542. So we also deleted the corresponding part in Table 1. And we also added the challenges of using blockchain into Discussion section, adding the references. Besides, we also considered and added the blockchain application in traceability system and cost control. Then, combining with our result, we discuss.

Comment 21:

‘References section has a huge quantity of conferences. This may harm the quality indicators of the journal. Perhaps, if the manuscript changes its composition it could be enhanced, reducing the conferences citation and enlarging the publication in scientific journals.’

Reply:

Because the structure of the paper has changed, the number of references and the position of the paper have also changed.

We reduced some conference papers -- 4, 8, 9, 17 in the previously submitted version. And added some journals papers-- 3, 5, 9, 10, 17, 21, 33, 34, 35. And we also changed the format of literature 24. And we have changed the position of reference [16] to replace original reference [2] which has been deleted.

……………………………………………

Thank you for your consideration of our manuscript.

Yours sincerely,

Dr. Xiong.

Round 2

Reviewer 1 Report

In this paper, a consortium blockchain-based agricultural machinery scheduling system is proposed for solving the problems of the traditional centralized scheduling system. This study seems useful, and interesting for many readers.

The items pointed out in the previous review are rewritten in detail, therefore almost acceptable. However, there remains some insufficient expressions as follows.

(1) As the explanations of Figure 12(a) are insufficient, Figure 12(a) is difficult to understand. For example, what “Time” and “Distance” of “Demand” mean? What is meant by “the blue line (Demand) is even higher than the red line (Real situation) at some moments”(line 519) ?

(2) In Section 4.1, the explanations for functions, etc. are strange, therefore difficult to understand. The following expressions should be reconsidered.

・Line 266:Min F →F, F(C) →C

・Line 269~272:

  Function(1) is the matching … →Function(1) (F) is the matching …

  While function(2) represent … →While function(2) (Tmij) represent …

      Function(3) is the cost … →Function(3) (C) is the cost …

・Line 267(Table 4):

  The distance between field I and j →The distance between field i and j

Author Response

Title: A Consortium Blockchain-Based Agricultural Machinery Scheduling System (Paper ID: sensors-774760)

Dear reviewer,

Sincerest thanks for your kind encouragement and constructive comments on our manuscript titled "A Consortium Blockchain-Based Agricultural Machinery Scheduling System" (paper ID: sensors-774760). We have thoroughly revised the paper in response to the insightful comments. We hope that all these modifications can meet your remarks. In the following, we will respond to detailed comments one by one. For easy reference, our replies to each of comments are highlighted in bold and blue fonts.

************************************

 ‘In this paper, a consortium blockchain-based agricultural machinery scheduling system is proposed for solving the problems of the traditional centralized scheduling system. This study seems useful, and interesting for many readers.

The items pointed out in the previous review are rewritten in detail, therefore almost acceptable. However, there remains some insufficient expressions as follows:’

Reply:

We thank you for your kind and insightful comments. And we have made some modifications according to the suggestions. In the following, each of your comments is replied accordingly.

Comment 1:

As the explanations of Figure 12(a) are insufficient, Figure 12(a) is difficult to understand. For example, what “Time” and “Distance” of “Demand” mean? What is meant by “the blue line (Demand) is even higher than the red line (Real situation) at some moments”(line 519) ?

Reply:

We thank you for your kind advice. We improved the simulation experiment in Round 1 modification, so we changed Figure 12 (a) at that time. In this round, we have further supplemented the interpretation of Figure 12 (a). Firstly, we explained the concept of Distance and Time in the figure on page 19 line 517-519, the revised sentences in the paper are as follows:

‘The y-coordinate is the distance that agricultural machinery should travel in the scheduling process, and the x-coordinate is the time taken for the agricultural machinery to travel a certain distance.’

Then, we supplemented the results in the figure. For the red line (Demand) and blue line (Real situation), we changed ‘the blue line (Demand) is even higher than the red line (Real situation) at some moments’ to ‘sometimes even above the red line’, and our modifications in the paper are as follows: (on page 19, line 520-526)

‘It can be seen that the blue line (Real situation) is close to the red line (Demand) and sometimes even above the red line, which means by using this system, in most cases, the time used for agricultural machinery scheduling is close to or even shorter than the time required by users. Among them, because the weather and road factors are also considered in the scheduling of the system, in order to be closer to the actual life situation, we add extreme weather and road conditions in the simulation experiment, that is, the position of x-axis close to 10h and 15h in the figure, which is also the reason why the blue line is so much below the red line.’

Comment 2:

In Section 4.1, the explanations for functions, etc. are strange, therefore difficult to understand. The following expressions should be reconsidered.

・Line 266:Min F →F, F(C) →C

・Line 269~272:

  Function(1) is the matching … →Function(1) (F) is the matching …

  While function(2) represent … →While function(2) (Tmij) represent …

      Function(3) is the cost … →Function(3) (C) is the cost …

・Line 267(Table 4):

The distance between field I and j →The distance between field i and j

Reply:

We thank you very much for your valuable comments in this regard. In Section 4.1, we have changed the expressions of function (1) and (3): changed ‘Min F’ to ‘F’, and changed ‘F(C)’ to ‘C’ . (page 8, line 266)

We also changed ‘Function(1) is the matching’ to ‘Function(1) (F) is the matching’, ‘While function(2) represent’ to ‘While function(2) (Tmij) represent’, and ‘Function(3) is the cost’ to ‘Function(3) (C) is the cost’. (page 9, line 269-272)

And in Table 4 (page 8, line 267), we changed ‘The distance between field I and j’ to ‘The distance between field i and j’.

……………………………………………

Thank you for your consideration of our manuscript.

Yours sincerely,

Dr. Xiong.

Reviewer 2 Report

The article has been considerably improved and most of the changes have been made.

Author Response

Title: A Consortium Blockchain-Based Agricultural Machinery Scheduling System (Paper ID: sensors-774760)

Dear reviewer,

Sincerest thanks for your kind encouragement and constructive comments on our manuscript titled "A Consortium Blockchain-Based Agricultural Machinery Scheduling System" (paper ID: sensors-774760). We have thoroughly revised the paper in response to the insightful comments. We hope that the modification can meet your remarks. In the following, we will respond to the detailed comment in bold and blue fonts.

************************************

 ‘The article has been considerably improved and most of the changes have been made.’.

Reply:

We are very grateful for your comments on the revision in the first round. It is very important for us to make the revision. We also have made some improvements to the article based on your valuable suggestions:

Figure 12(a) in Result section has been supplemented to make it easier to understand. We made modifications and additions in line 517-519 and line 520-526.

……………………………………………

Thank you for your consideration of our manuscript.

Yours sincerely,

Dr. Xiong.
